# Prenatal and postnatal challenges affect the hypothalamic molecular pathways that regulate hormonal levels

Sandra L. Rodriguez-Zas[1,2,3,4,5]*, Nicole L. Southey[6], Laurie Rund[1], Adrienne M. Antonson[1,3,5], Romana A. Nowak[1,5], Rodney W. Johnson[1,2,3]

1 Department of Animal Sciences, University of Illinois at Urbana-Champaign, Urbana, IL, United States of America, 2 Division of Nutritional Sciences, University of Illinois at Urbana-Champaign, Urbana, IL, United States of America, 3 Neuroscience Program, University of Illinois at Urbana-Champaign, Urbana, IL, United States of America, 4 Department of Statistics, University of Illinois at Urbana-Champaign, Urbana, IL, United States of America, 5 Carl R. Woese Institute for Genomic Biology, University of Illinois at Urbana-Champaign, Urbana, IL, United States of America, 6 Department of Bioengineering, University of Illinois at Urbana-Champaign, Urbana, IL, United States of America

* rodrgzzs@illinois.edu

**Data Availability Statement:** The data is located in the National Center for Bioinformatics Institute (NCBI), Gene Expression Omnibus (GEO

## Abstract

This study aimed to improve our understanding of how the hypothalamus mediates the effects of prenatal and postnatal challenges on behavior and sensitivity to stimuli. A pig model of virally initiated maternal immune activation (MIA) was used to investigate potential interactions of the prenatal challenge both with sex and with postnatal nursing withdrawal. The hypothalami of 72 females and males were profiled for the effects of MIA and nursing withdrawal using RNA-sequencing. Significant differential expression (FDR-adjusted p value < 0.05) was detected in the profile of 222 genes. Genes involved in the Gene Ontology biological process of regulation of hormone levels tended to be over-expressed in individuals exposed to both challenges relative to individuals exposed to either one challenge, and most of these genes were over-expressed in MIA females relative to males across nursing levels. Differentially expressed genes included *Fshb*, *Ttr*, *Agrp*, *Gata3*, *Foxa2*, *Tfap2b*, *Gh1*, *En2*, *Cga*, *Msx1*, and *Npy*. The study also found that prenatal and postnatal challenges, as well as sex, impacted the regulation of neurotransmitter activity and immune effector processes in the hypothalamus. In particular, the olfactory transduction pathway genes were over-expressed in weaned MIA males, and several transcription factors were potentially found to target the differentially expressed genes. Overall, these results highlight how multiple environmental challenges can interact and affect the molecular mechanisms of the hypothalamus, including hormonal, immune response, and neurotransmitter processes.

## Introduction

Inflammatory signals elicited by infection or other immune challenges can alter molecular mechanisms in the brain associated with behavioral disorders and pain sensitivity. Maternal immune activation (MIA) during gestation is associated with neurodevelopmental pathologies

repository, experiment identifier GSE227725. The secure token to allow review is cdyxkkoidjmhhst.

**Funding:** This research was funded by United States Department of Agriculture (USDA) National Institute of Food and Agriculture (NIFA), Agriculture and Food Research Initiative (AFRI) award number 2018-67015-27413, USDA NIFA University of Illinois at Urbana-Champaign Experiment Station (ILLU) award number 2022-38420-38610, and NIH NIDA award number P30 DA018310. Role the funders: The funders had no role in study design, data collection and analysis, decision to publish, or preparation of the manuscript.

**Competing interests:** The authors have declared that no competing interests exist.

in the offspring, including schizophrenia and autism spectrum disorders and sensory processing sensitivity. In rodent and pig models of MIA, offspring display an increased incidence of autism and schizophrenia-like phenotypes, social deficit, and modified activity levels [1–4]. Likewise, postnatal immune challenges, including inflammatory agents and environmental stressors, can cause behavioral and allodynia alterations [5, 6].

Infection and environmental stressors can alter the balance of inflammatory factors impacting the hypothalamic mechanisms that modulate the pituitary and adrenal gland function in the hypothalamic-pituitary-adrenal (HPA) axis [7]. The impact of the previous factors can have multiple consequences because the hypothalamus regulates various processes, including the endocrine and autonomic nervous systems, hormone release and activity cycles, circadian rhythm, and feeding and social behaviors. In turn, the hypothalamus also participates in the modulation of stress hormones and the response to stressors.

Hyperactivity of the HPA axis, typically detected by high peripheral cortisol, has been associated with MIA-related behavioral disorders, including autism and schizophrenia spectrum disorder, anxiety, depression, hypersensitivity to pain, hyperalgesia, and allodynia [7, 8]. The link between the hypothalamic and behavioral disruption was reported in NIH Swiss mice exposed to PolyI:C-induced MIA, which presented behavioral disruptions and lower levels of hypothalamic vasopressin receptor 1a (Avpr1a) gene expression [9]. Also, P85 male mice exposed to PolyI:C-induced MIA displayed increased pain sensitivity and hypothalamic over-expression of the KiSS-1 metastasis suppressor (Kiss1) gene, while no significant changes were detected in females [7].

Postnatal maternal care has long-lasting effects on offspring development in the biomedical rodent and pig models. The stress elicited by weaning offspring, including withdrawal from nursing, the mother, and littermates, can impact the HPA axis activity. Weaning stress increased the peripheral cortisol level and lowered the hypothalamic expression of the corticotropin releasing hormone (Crh) gene in 28-day-old male pigs [10]. Also, early weaning at P21 (relative to P28) increased the hypothalamic density of the oxytocin receptor in male but not in female mice [11].

Studies about the impact of MIA or nursing withdrawal on the hypothalamus have focused on one of the challenges and profiled genes and gene products that had established relationships with stress or inflammatory responses [7, 12]. This study aimed to understand the simultaneous impact of prenatal and postnatal challenges in the transcriptome of the hypothalamus within and across sexes, considering the multiple signals and physiological and behavioral processes regulated by this structure. Gene pathway reconstruction advanced the understanding of the simultaneous effects of MIA and a second stress on the interconnected molecular mechanisms underlying the processes modulated by the hypothalamus.

## Materials and methods

Published protocols were followed for all experimental procedures [13]. Animal studies were approved by the Illinois Institutional Animal Care and Use Committee (IACUC) and abide by the United States Department of Agriculture Animal Welfare Act and the National Institutes of Health (NIH) Public Health Service Policy on the Humane Care and Use of Animals. Gestating female pigs (Camborough breed) were separately housed in disease containment chambers, exposed to a 12-hour light/dark cycle, and fed a diet that met gestational requirements and ad libitum water availability [14]. Porcine respiratory and reproductive virus (PRRSV; strain P129-BV) diluted in sterile Dulbecco's modified Eagle medium was inoculated to eight gilts, offering MIA challenge to the developing offspring. Another set of eight gilts was inoculated with the medium and was studied as a control. Infection status was confirmed 7 days

after inoculation using a PCR test. Virally inoculated gilts had lower feed consumption and higher body temperature for 14 days relative to control gilts, and feed availability was adjusted to match intake between gilt groups [1].

On gestation day 113, farrowing was induced, and all pigs were nursed and remained with their mothers fed a diet that satisfied lactation requirements [13]. At three weeks of age, approximately half of each litter continued nursing, and the remaining pigs were withdrawn from the litter and nursing. The weaned pigs were housed in groups of four per room and received a diet that satisfied growing requirements and ad libitum water, and the experiment concluded one day later.

Altogether, 72 females and males from two prenatal inflammation groups (MIA and control) assigned to one of two postnatal challenges (nursing and nursing withdrawn) were studied. A diagram depicting the experimental design is available in S1 Fig. The pigs were anesthetized using a dose of 0.03 mL/kg body weight of 4.4 mg/kg Telazol (100 mg/mL, Zoetis, Parsippany, NJ, USA), 2.5 mL of ketamine (100 g/L) and 2.5 mL of xylazine (100 g/L) [15]. The anesthesia was applied intramuscularly, followed by an intracardiac injection of Fatal Plus (Vortech Pharmaceuticals, MI, USA) sodium pentobarbital (86 mg/kg body weight). The hypothalamus of the 22-day-old pigs was extracted, flash-frozen on dry ice, and maintained at $-80°C$ using published protocols [16]. The hypothalamic RNA was isolated with the EZNA isolation kit (Omega Biotek, Norcross, GA, USA) per the manufacturer's instructions. All samples reached an RNA integrity number above 7.4, which indicates low degradation. The Roy J. Carver Biotechnology Center (Urbana, IL, USA) processed the RNA-seq libraries using TruSeq Stranded mRNAseq Sample Prep kit (Illumina Inc, San Diego, CA, USA), which were subsequently quantified by qPCR, and the fragments were sequenced from both ends on two lanes on a NovaSeq 6000 sequencer for 151 cycles using NovaSeq 6000 reagent kit. The sequences were demultiplexed and compiled into FASTQ files using the bcl2fastq v2.20 routine. The average and median number of reads per sample were 259.9 million and 254.5 million, respectively. The experiment identifier GSE227725 in the public Gene Expression Omnibus repository includes the raw sequencing files corresponding to the 72 samples [17].

Read trimming was unnecessary because all read positions' quality was above a Phred score $> 36$. The *Sus scrofa* transcriptome v. 11.1 was used as the reference to align the paired-end reads, and transcript levels were quantified using kallisto v. 0.43.0 and default specifications. The levels of the gene expression were measured as sequence reads couns and were adjusted for library size and gene length. Genes with fewer than 5 reads and lower that 1 adjusted count per pig group were removed from subsequent analysis. The logarithm base 2 ransformed gene expression values were analyzed using a generalized linear model to test the effects of prenatal inflammation, nursing withdrawal, sex, and interactions on 18,295 genes analysis using the software edgeR v. 3.14.0 in the R v. 3.3.3 environment. The p values are adjusted for multiple testing using the False Discovery Rate (FDR) method [18].

Over-representation of gene ontology (GO) biological processes (BP) [19] and Kyoto Encyclopedia of Genes and Genomes (KEGG) pathways [20] were studied [21]. *Sus scrofa* was the reference genome used, and the enrichment analyses were performed using the hypergeometric over-representation analysis (ORA) [22] and the gene set enrichment analysis (GSEA) [16, 23]. While the ORA analysis considered genes differentially expressed at FDR p-value $< 0.1$, GSEA accounted for the $\log_2$(fold change) between factor groups to detect enrichment among the over- or under-expressed genes. The ORA enrichment ratio (ER), GSEA normalized enrichment score (NES), and FDR-adjusted p-value (based on 1000 permutations) were considered in identifying enriched functional categories.

Further understanding of the effects of MIA and nursing withdrawal on the hypothalamus molecular pathways was gathered by integrating gene coexpression networks with gene

differential expression profiles. The relationships among genes annotated to statistical and biologically significant enriched categories were obtained from the STRING database v. 11.5 of functional associations [24] and visualized in the Cytoscape v. 3.9.1 platform [25] following published approaches [16]. The nodes represent genes, and the edges represent the protein relationships in STRING experimental, coexpression, text mining, and repositories [24]. The red-to-yellow-to-green color scheme of the nodes denotes the transition from under-expression to equal and over-expression in the first treatment group relative to the second treatment group in the comparison portrayed by the network. While there are no protein profiles to confirm the expression and co-expression patterns, statistical robust findings rely on the consideration of orthogonal contrasts among the 72 pigs distributed across 8 groups application of an stringent threshold of FDR-adjusted p-value < 0.05 for differential expression, results and interpretation focus on consistency across categories, approaches, and benchmarked against published annotation.

Regulatory mechanisms shared by target genes presenting differential expression associated with MIA, nursing withdrawal, and sex were detected using the iRegulon software v. 1.3, build 1024 [26]. This analysis identified enriched regulatory transcription factor binding motifs and genomic tracks using the area under the cumulative recovery curve (AUC). The AUCs, estimated in the top 3% of the homotypic motif clusters across 10 vertebrate species, were standardized in the iRegulon NES. The NES > 3 threshold that corresponds to a 0.03 < FDR p-value < 0.09 were applied in the present study [26, 27].

## Results

Among the 18,295 genes analyzed, at FDR-adjusted p value < 0.05, significant effects were identified on 39 genes for virally induced MIA, 16 genes for nursing withdrawal, 61 genes for sex, 27 genes for MIA-by-nursing withdrawal, 42 genes for MIA-by-sex, and 37 genes for sex-by-nursing withdrawal effect (**S1 Table**). Focusing on the genes that were impacted by the factors studied, **Fig 1** depicts the distribution of the differential expression ($\log_2$(fold change)) among the genes presenting significant (FDR-adjusted p value < 0.1) main effects of MIA, nursing withdrawal, or sex, divided into over- and under-expressed genes within effect. The majority of the genes impacted by the factors studied were over-expressed in MIA relative to control, over-expressed in withdrawal relative to nursing, and in males relative to females.

### Hypothalamic biological processes and genes networks associated with challenges and sex effects

The enriched GO biological processes including at least 10 genes differentially expressed (FDR-adjusted p value < 0.05) across MIA, nursing withdrawal, sex, and interactions effects identified by ORA, are presented in **Table 1**. An extended list of biological processes including at least 5 genes differentially expressed (FDR-adjusted p value < 0.05) across the factors studied is available in **S2 Table**. The enriched biological processes share endogenous signaling mechanisms, where G protein-coupled receptors are membrane receptors of steroid hormones. Estrogen receptors link estrogen signaling to promoter-related RNA polymerase II and transcription mechanisms, evidenced in the detected positive regulation transcription by RNA polymerase II category (GO:0045944) (Table 1).

The $\log_2$(fold change) of genes differentially expressed (FDR-adjusted p value < 0.05) for the effects of MIA, nursing withdrawal, sex, and interactions annotated to the enriched GO biological processes (Table 1) are summarized in **Table 2**. The differentially expressed genes include follicle-stimulating hormone beta polypeptide (*Fshb*), transthyretin (*Ttr*), agouti-related neuropeptide (*Agrp*), GATA binding protein 3 (*Gata3*), forkhead box A4 (*Foxa2*),

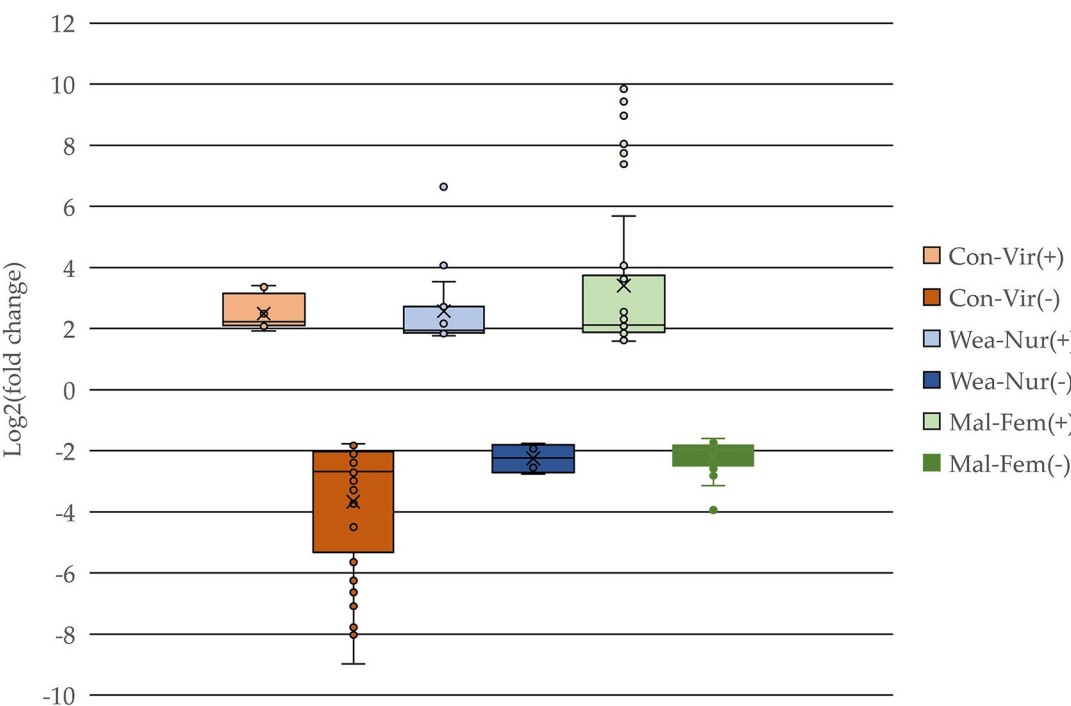

**Fig 1. Boxplots of the differential expression (log₂(fold change)) of the genes presenting significant effects (False Discovery Rate-adjusted p value < 0.1) of maternal immune activation (red: Con-Vir = Control versus Viral prenatal challenge), nursing withdrawal (blue: Wea-Nur = Weaning versus Nursing), or sex (grey: Mal-Fem = Male versus Female), partitioned by differential expression sign.**

transcription factor AP-2 beta (*Tfap2b*), growth hormone 1 (*Gh1*), glycoprotein hormones, engrailed homeobox 2 (*En2*), alpha polypeptide (*Cga*), MSH homeobox 1 (*Msx1*), neuropeptide Y (*Npy*), neuromedin U (*Nmu*), prolactin (*Prl*), POU class 4 homeobox 1 (*Pou4f1*), nuclear receptor subfamily 5 group A member 2 (*Nr5a2*), paired box 3 (*Pax3*), KiSS-1 metastasis-suppressor (*Kiss1*), thyroid stimulating hormone receptor (*Tshr*), luteinizing hormone beta polypeptide (*Lhb*), aldehyde dehydrogenase 1 family member A1(*Aldh1a1*), C-X-C motif chemokine ligand 9 (*Cxcl9*), and eukaryotic translation initiation factor 1A, Y-linked (*Eif1ay*). Significant interactions between MIA and the remaining factors indicate that the impact of MIA on gene expression depends on the nursing status and sex of the individual. Likewise, a significant interaction between nursing withdrawal and sex indicates that the impact of such stressor depends on the sex of the individual.

**Table 1. Enriched Gene Ontology (GO) biological processes, including at least 10 genes differentially expressed (False Discovery Rate FDR-adjusted p value < 0.05) in response to maternal immune activation, nursing withdrawal, sex, or interactions.**

| GO Identifier | Description | Gene count | Enrichment [1] | p value | FDR p value |
|---|---|---|---|---|---|
| GO:0010817 | regulation of hormone levels | 11 | 7.9 | $8.8 \times 10^{-08}$ | $7.9 \times 10^{-04}$ |
| GO:0010469 | regulation of signaling receptor activity | 10 | 6.7 | $1.6 \times 10^{-06}$ | $5.5 \times 10^{-03}$ |
| GO:0071495 | cellular response to endogenous stimulus | 13 | 3.5 | $3.7 \times 10^{-05}$ | $5.6 \times 10^{-02}$ |
| GO:0045944 | positive regulation transcription by RNA polymerase II | 11 | 3.4 | $2.1 \times 10^{-04}$ | $1.2 \times 10^{-01}$ |
| GO:0007186 | G protein-coupled receptor signaling pathway | 11 | 3.1 | $4.9 \times 10^{-04}$ | $1.8 \times 10^{-01}$ |

[1] Over-representation analysis enrichment ratio.

**Table 2. Log$_2$(fold change) of the differentially (False Discovery Rate FDR-adjusted p value < 0.05) expressed genes (#) for the effects of maternal immune activation (MIA), nursing withdrawal (Nur), sex, and interactions (-) annotated to enriched Gene Ontology biological processes (BP).**

| Gene | Enriched BP[1] | MIA-Nur | Nur-Sex | Sex-MIA | MIA | Nur | Sex |
|---|---|---|---|---|---|---|---|
| *Agrp* | R,S,G | -2.86# | -3.79# | -2.43 | -2.98# | 2.71# | 2.54# |
| *Aldh1a1* | H | 0.26 | 3.03# | 1.46 | 0.41 | -1.41 | -0.95 |
| *Cga* | H,R,T,G | -5.66# | -3.86# | -2.36 | -6.64# | 0.02 | 3.62# |
| *Cxcl9* | R,G | 1.81 | 4.78# | 2.87 | -1.84 | -1.26 | -1.08 |
| *En2* | T | -3.22# | 6.25# | 0.14 | -2.40# | -1.27 | -2.59# |
| *Foxa2* | H,T | -1.59 | 4.43# | 2.17 | 1.01 | -0.52 | -3.94# |
| *Fshb* | H,R,S,T,G | -3.48# | -0.94 | 0.53 | -3.25# | 0.95 | 0.85 |
| *Gata3* | H,S,T | -2.64# | 2.13 | 2.00 | -0.33 | 0.26 | -2.46# |
| *Gh1* | R,S | 0.50 | 3.87# | 5.99# | 0.49 | -0.76 | -2.61# |
| *Gnrh2* | R,G | 1.14 | 1.45 | 1.96 | 1.91 | -1.45 | -1.96# |
| *Kiss1* | H,G | -2.20 | -2.71# | -2.52 | -2.65# | 1.50 | 1.77# |
| *Lhb* | H,R,S,G | -2.06 | -0.17 | 1.25 | -2.34# | -0.70 | 0.48 |
| *Msx1* | S,T | -1.57 | -0.61 | 0.36 | -1.98# | -0.16 | 0.42 |
| *Nmu* | H,G | -1.76 | -2.91# | -1.52 | -1.76 | 1.44 | 1.69 |
| *Npy* | R,G | -2.21 | -2.86# | -2.01 | -2.10# | 2.31# | 1.84# |
| *Nr5a2* | S,T | -2.10 | -2.84# | -0.94 | -1.15 | 1.51 | 1.45 |
| *Pax3* | T | -0.85 | 1.54 | 2.71# | 0.93 | -0.04 | -1.78# |
| *Pou4f1* | S,T | -3.39# | 5.37# | 2.84# | -0.57 | -0.66 | -3.15# |
| *Prl* | R,S,T | 1.71 | 0.86 | 6.16# | -0.07 | 0.03 | -0.25 |
| *Tfap2b* | H,S,T | -1.15 | 0.49 | 3.11# | 0.68 | 0.34 | -1.55 |
| *Tshr* | S,G | -1.13 | -2.54# | -0.86 | -1.09 | 0.92 | 1.49 |
| *Ttr* | H,R | 3.32# | -0.65 | -0.63 | 1.80 | -1.23 | 1.53 |

[1] Enriched biological process: H = regulation of hormone levels; R = regulation of signaling receptor activity; S = cellular response to endogenous stimulus; T = positive regulation of transcription by RNA polymerase II; G = G protein-coupled receptor signaling pathway.

[2] Sign of log$_2$(fold change) for main effects: +(-) over- (under)-expression in control relative to MIA group, nursing withdrawal relative to the nursing group, or males relative to females; for interactions: +(-) over- (under-) expression in withdrawn nursing males, withdrawn nursing control, and control males relative to other groups.

Most genes in the enriched endocrine signaling processes (Table 1) presented a negative log$_2$(fold change) for the MIA-by-nursing interaction indicating. This profile suggests that genes tended to be over-expressed in individuals exposed to both challenges (MIA and nursing withdrawal) relative to individuals exposed to either one challenge across the sexes (Table 2). A positive log$_2$(fold change) trend was observed for the sex-by-MIA interaction, indicating that most genes were over-expressed in MIA females relative to MIA males across nursing levels. While the distribution of log$_2$(fold change) signs was even for the nursing-by-sex interaction, the most extreme values had a positive sign indicating that these genes were over-expressed in nursing-withdrawn males relative to females.

Considering the main effect of MIA among the genes in Table 2, the most differentially expressed genes had a negative log$_2$(fold change), which marks predominant over-expression in MIA relative to control pigs. Considering the main effect of sex, most extreme log$_2$(fold change) had a negative sign corresponding to over-expression in females relative to males. The positive log$_2$(fold change) of two genes presenting extreme postnatal stressor effect corresponds to over-expression in nursing withdrawn relative to individuals that remained with the mother nursing (Table 2).

The combined study of the expression profiles across MIA-nursing-sex groups and molecular interactions among the genes in the regulation of hormone levels process (GO:0010817) enabled additional characterization of the hypothalamic dysregulation. The visualization of

the network of genes in the hormonal regulation pathway was selected for visualization across 12 contrasts because of the significant enrichment, and also because it includes genes from other enriched pathways including regulation of transcription by RNA polymerase II, G-protein-coupled receptor signaling pathway, cellular response to endogenous stimulus, regulation of signaling receptor activity. Fig 2 depicts the relationship between genes in the regulation of hormone levels process in the STRING database while comparing groups that differ in the MIA level or sex within a nursing group. Fig 3 depicts the relationship between genes in the regulation of hormone levels process while comparing groups that differ in the nursing status or sex within a MIA level. The genes annotated to the process regulation of hormone levels include *Aldh1a1*, *Cga*, *Foxa2*, *Fshb*, *Gata3*, *Kiss1*, *Lhb*, *Nmu*, *Tfap2b*, and *Ttr*. The gene Esr1 was added to the network to facilitate the depiction of connections between the genes in the process. The color scheme ranges from red or under-expressed in the first group, to yellow (equally expressed), and green or over-expressed in the first group relative to the second group of the contrast. The $\log_2$(fold changes) represented by the color scheme are listed in **S4 Table**.

**Fig 2** networks c and d depict a predominance of genes over-expressed in nursed MIA relative to control pigs across sexes, whereas the profile tended to revert in nursing withdrawn pigs (Fig 2A and 2B). **Fig 3** networks a and b depict a predominance of genes under-expressed in MIA withdrawn relative to nursing pigs, whereas the profile became less extreme and reverted in control pigs (Fig 3C and 3D). Notably, the gene expression differences between sexes were more extreme in pigs not exposed to challenges (Fig 2F) and weaker in pigs exposed to both challenges (Fig 2E). Gene estrogen receptor 1 (*Esr1*) is a hub gene presenting the highest network radiality, closeness centrality, and betweenness centrality among all genes in the network. As a hub, gene *Esr1* presented weak changes in gene expression across prenatal or postnatal challenges or sexes.

## Gene set and regulatory enrichment analysis of differentially expressed genes in the hypothalamus

Enrichment results from the GSEA approach enabled the characterization of the MIA, nursing, and sex effects on KEGG pathways and GO biological processes among genes presenting extreme differential expression. Table 3 summarizes the KEGG pathways and GO biological processes enriched at FDR-adjusted p value < 0.05 in at least two experimental factors. An extended list of all enriched KEGG pathways and GO biological processes identified by GSEA at FDR-adjusted p value < 0.05 is available in **S3 Table**.

The predominance of inflammatory response and neurotransmitter functional categories in Table 3 highlights the complementary understanding of the molecular mechanisms impacted by MIA, nursing withdrawal, and sex offered by GSEA. Inflammatory response categories associated with the virally induced MIA studied include the viral diseases pathways Influenza A (ssc05164) and Herpes simplex infection (ssc05168) and the GO immune effector process (GO:0002252) and reactive nitrogen species metabolic process (GO:2001057). Examples of enriched neuronal signaling categories include the cell adhesion molecules (ssc04514) and regulation of neurotransmitter levels (GO:0001505).

The sign of the NES is consistent across enriched inflammatory response and neurotransmitter categories and indicates over-expression in nursed MIA females relative to other groups. The consistent profile of the mitochondrial transport process (GO:0006839) corresponds to neuroinflammation impairment of axonal mitochondrial transportation. Contrary to the previous trend, the NES sign of the olfactory transduction pathway (ssc04740) indicates that the genes in this pathway were predominantly over-expressed in nursing-withdrawn, control males.

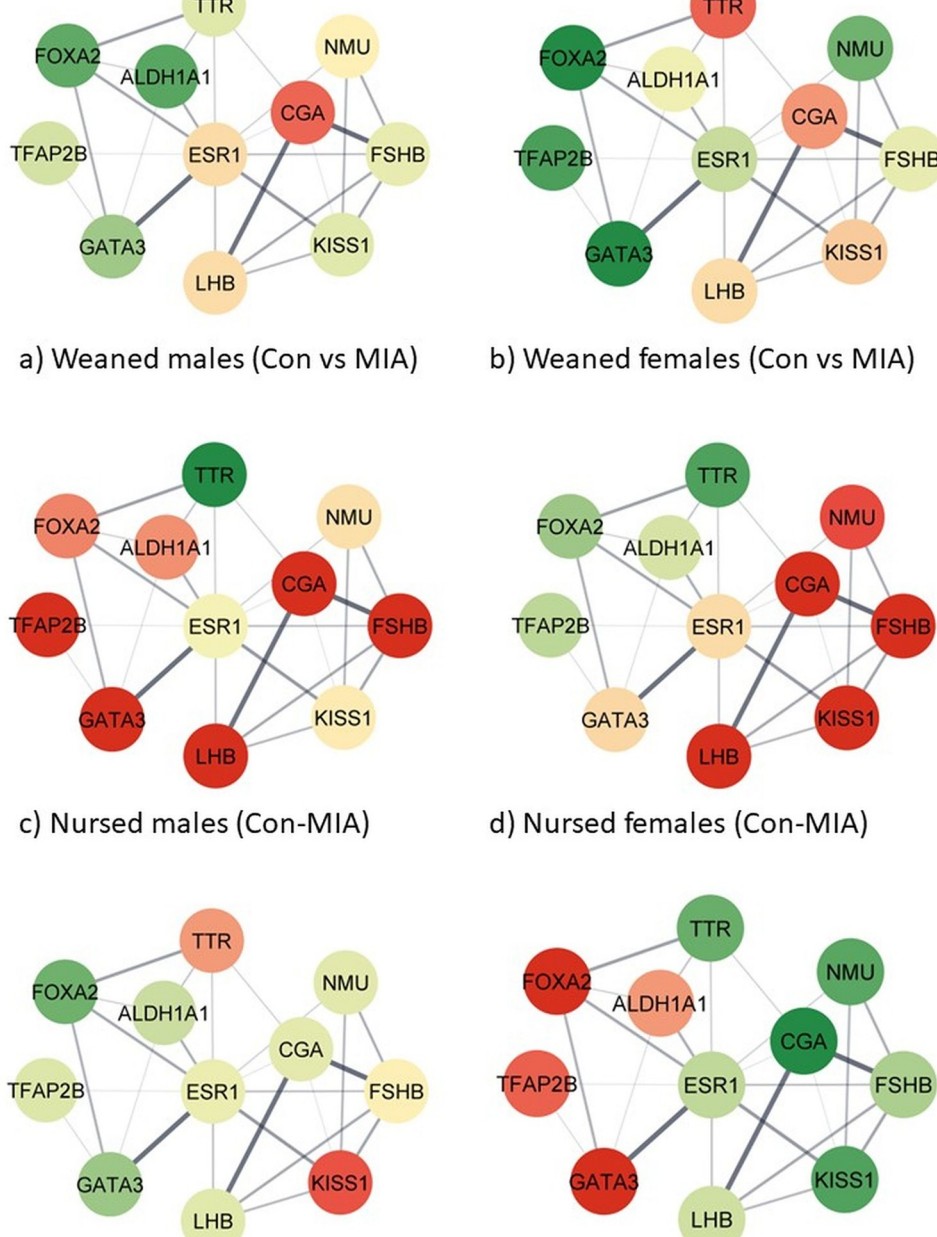

a) Weaned males (Con vs MIA)  b) Weaned females (Con vs MIA)

c) Nursed males (Con-MIA)  d) Nursed females (Con-MIA)

e) MIA weaned (males vs females)  f) Con nursed (males vs females)

**Fig 2. Relationship and expression profiles of genes annotated to the enriched (FDR-adjusted p value < 0.0008) Gene Ontology biological process regulation of hormone levels (GO:0010817) between viral maternal immune activation (MIA) and control (Con) groups and between males and females under withdrawal from nursing (weaned) and nursed) conditions where the color scheme red-to-yellow-to-green denotes under->no->over-expression in the first relative to the second in the specified contrast (-).**

The analysis of regulatory motifs and genomic tracks encompassing transcription factor binding sites among the differentially expressed genes (FDR-adjusted p value < 0.05) highlighted shared modulatory mechanisms. Overall, 71 motifs and 9 tracks annotated to 57 transcription factors were enriched at approximate FDR-adjusted p-value < 0.05, normalized

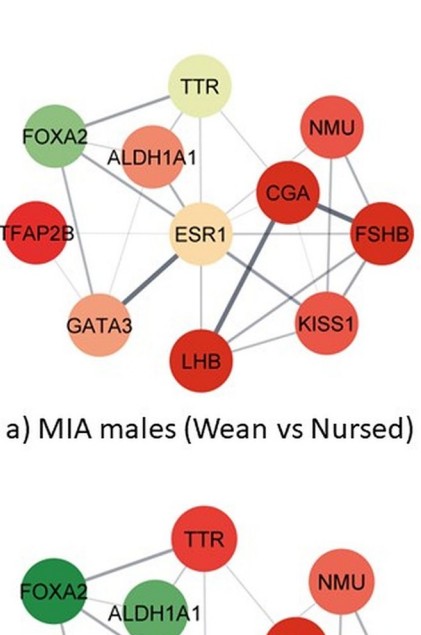
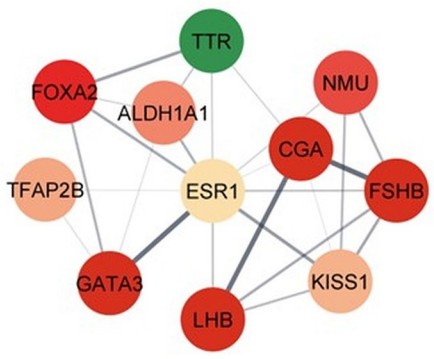

a) MIA males (Wean vs Nursed)          b) MIA females (Wean vs Nursed)

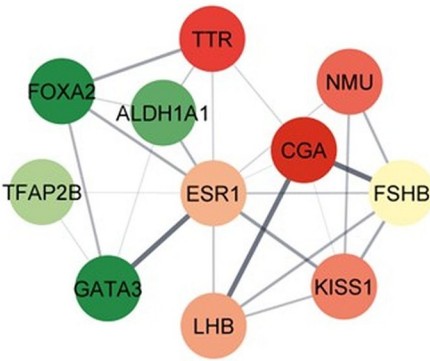
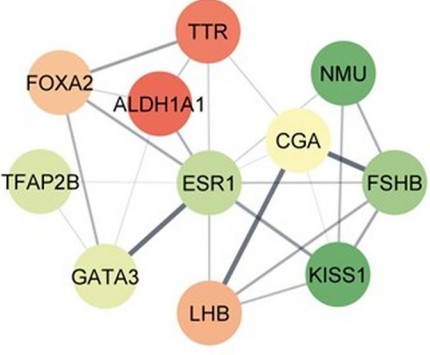
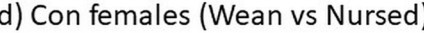

c) Con males (Wean vs Nursed)          d) Con females (Wean vs Nursed)

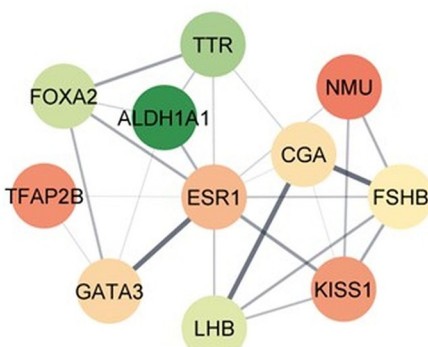
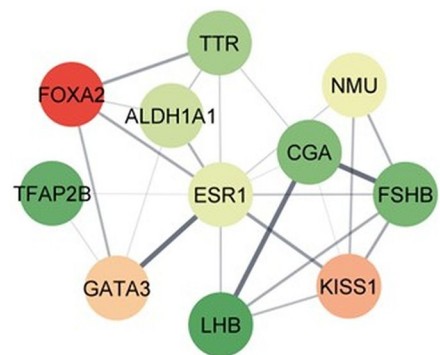

e) Con weaned (males vs females)       f) MIA nursed (males vs females)

**Fig 3. Relationship and expression profiles of genes annotated to the enriched (FDR-adjusted p value < 0.0008) Gene Ontology biological process regulation of hormone levels (GO:0010817) between withdrawal from nursing (weaned) and nursed groups and between males and females under viral maternal immune activation (MIA) and control (Con) conditions where the color scheme red-to-yellow-to-green denotes under->no->over-expression in the first relative to the second in the specified contrast (-).**

enrichment score > 3 among the target genes impacted by the effects of MIA, nursing withdrawal, and sex. **Table 4** lists the motifs or tracks for the transcription factors presenting normalized enrichment score > 4. An extended list of motifs, tracks, and transcription factors is available in **S5 Table**. Enriched transcription factors listed in Table 4 and **Fig 4** include ALX

**Table 3. Normalized enrichment score of KEGG pathways and GO biological processes (BP) enriched at False Discovery Rate FDR-adjusted p value < 0.05 among genes with profiles impacted by at least two experimental factors including maternal immune activation (MIA), nursing (Nur), sex, or interactions (-).**

| KEGG or BP | Description | MIA-Nur | Nur-Sex | MIA-Sex | MIA | Nur | Sex |
|---|---|---|---|---|---|---|---|
| Pathway | | | | | | | |
| ssc05330 | Allograft rejection | | 2.49 | | -2.28 | -2.29 | -2.16 |
| ssc04612 | Antigen processing and presentation | 2.95 | 2.64 | 2.67 | | -2.79 | -2.45 |
| ssc04514 | Cell adhesion molecules (CAMs) | 2.31 | | 2.44 | | | |
| ssc00260 | Glycine, serine and threonine metabolism | | 2.25 | | | | -1.82 |
| ssc05332 | Graft-versus-host disease | | 2.36 | | -2.14 | -2.31 | -2.26 |
| ssc05168 | Herpes simplex infection | | | | | -2.17 | -1.94 |
| ssc05164 | Influenza A | 2.30 | | 2.49 | | | |
| ssc04672 | Intestinal immune network for IgA production | 2.47 | 2.23 | 2.46 | -1.99 | | |
| ssc05140 | Leishmaniasis | 2.29 | 2.33 | 2.41 | | | |
| ssc04740 | Olfactory transduction | -2.28 | -2.07 | -2.59 | | 2.17 | 2.39 |
| ssc05012 | Parkinson disease | 2.44 | | | | -2.67 | |
| ssc04145 | Phagosome | 2.32 | | 2.44 | | | |
| ssc03010 | Ribosome | | 2.28 | | -2.11 | -2.77 | |
| ssc05150 | Staphylococcus aureus infection | 2.96 | | 2.60 | -2.16 | -2.59 | -1.88 |
| ssc04658 | Th1 and Th2 cell differentiation | | | 2.39 | | | -1.87 |
| ssc04940 | Type I diabetes mellitus | 2.69 | 2.32 | 2.41 | -2.12 | -2.28 | -2.12 |
| ssc05416 | Viral myocarditis | | 2.17 | | -2.16 | | -1.81 |
| Process | | | | | | | |
| GO:0051186 | cofactor metabolic process | 1.97 | | | | -1.82 | |
| GO:0002252 | immune effector process | | | 2.18 | | -1.93 | |
| GO:0006839 | mitochondrial transport | 2.13 | | | | -2.36 | |
| GO:0006836 | neurotransmitter transport | 2.41 | | | | -1.81 | |
| GO:2001057 | reactive nitrogen species metabolic process | 2.20 | | 2.03 | | | |
| GO:0031099 | regeneration | 1.89 | | | | -2.16 | |
| GO:0001505 | regulation of neurotransmitter levels | 2.54 | | 2.16 | | | |

[2] Sign of the normalized enrichment score NES for main effects: +(-) over- (under)-expression in control relative to MIA group, nursing withdrawal relative to the nursing group, or males relative to females; for interactions: +(-) over- (under-) expression in withdrawn nursing males, withdrawn nursing control, and control males relative to other groups.

**Table 4. Top enriched motif or genomic track (normalized enrichment score NES > 4) within transcription factor and the number of differentially expressed target genes (False Discovery Rate-adjusted p value < 0.05) for the effects of maternal immune activation, nursing withdrawal, sex, and interactions.**

| Transcription factor(s) [1] | Motif/track | AUC[2] | NES | Genes |
|---|---|---|---|---|
| SUZ12/*FOXA2* | wgEncodeSydhTfbsNt2d1Suz12UcdPk.narrowPeak.gz | 0.18 | 10.65 | 17 |
| CTBP2/PAX3 | wgEncodeSydhTfbsH1hescCtbp2UcdPk.narrowPeak.gz | 0.16 | 9.25 | 13 |
| TCF12/*FOXA2* | wgEncodeHaibTfbsA549Tcf12V0422111Etoh02PkRep2.broadPeak.gz | 0.09 | 4.81 | 11 |
| PROP1 | taipale-TAATYNAATTA-PROP1-DBD | 0.13 | 4.81 | 32 |
| ALX4 | homer-M00250 | 0.12 | 4.29 | 45 |
| FOXB1 | taipale-WNWGTMAATA*TTR*ACWNW-FOXB1-DBD | 0.12 | 4.24 | 11 |
| SIRT6 | transfac_pro-M01797 | 0.12 | 4.23 | 11 |

[1] Regulatory region or track in the iRegulon database encompassing one or multiple genes.

[2] Area Under the cumulative Recovery Curve.

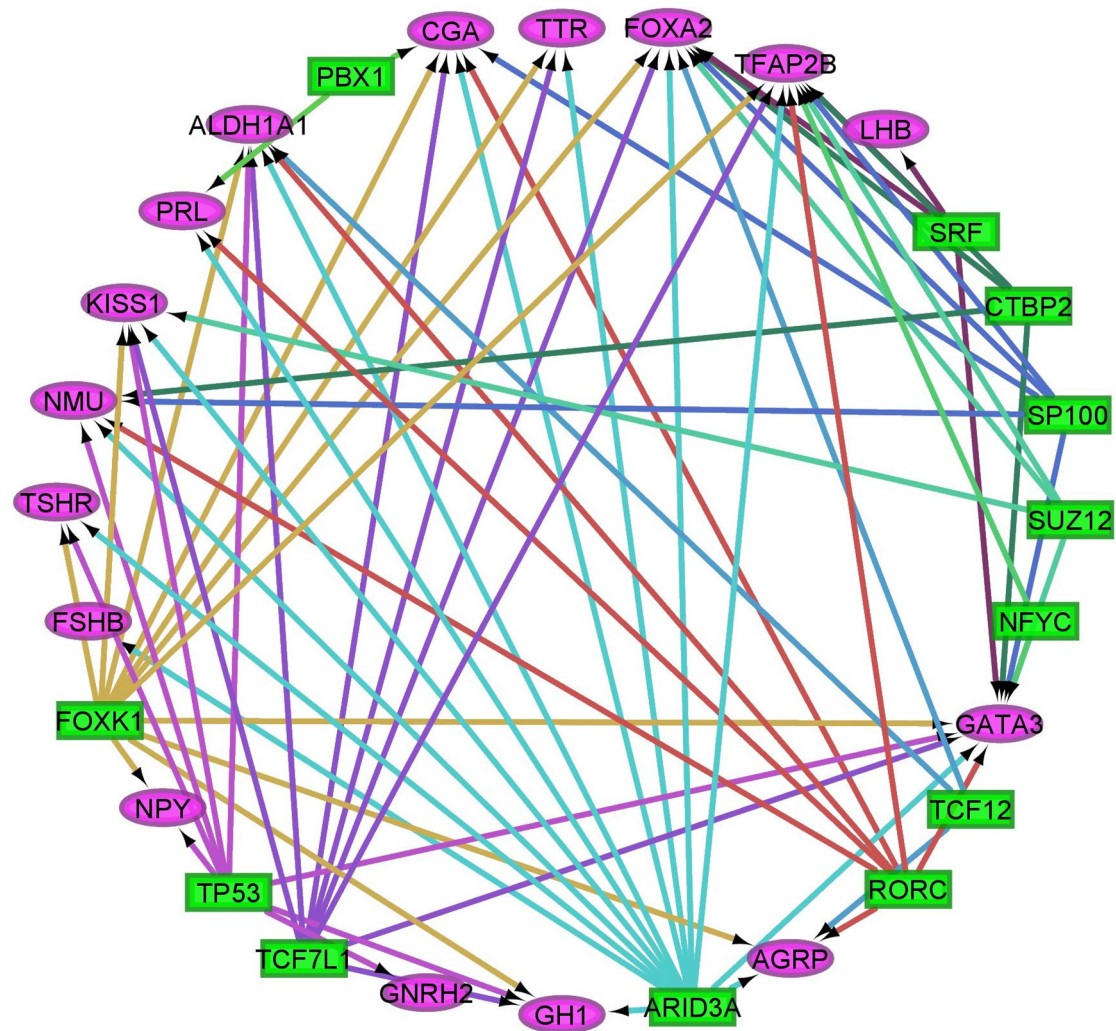

**Fig 4. Relationship between transcription factors (rectangles) and differentially expressed target genes (ellipses) annotated to the enriched (False Discovery Rate FDR-adjusted p value < 0.001) Gene Ontology biological process regulation of hormone levels (GO:0010817).**

homeobox 4 (ALX4), C-terminal binding protein 2 (CTBP2), FOXA2, forkhead box B1 (FOXB1), PROP paired-like homeobox 1 (PROP1), sirtuin 6 (SIRT6), SUZ12 polycomb repressive complex 2 subunit (SUZ12), transcription factor 12 (TCF12), AT-rich interaction domain 3A (ARID3A), forkhead box K1 (FOXK1), nuclear transcription factor Y subunit gamma (NFYC), PBX homeobox 1 (PBX1), paired box 3 (PAX3), RAR related orphan receptor C (RORC), SP100 nuclear antigen (SP100), serum response factor (SRF), transcription factor 7 like 1 (TCF7L1), and tumor protein p53 (TP53).

A bimodality can be observed among transcription factors with target genes in the process regulation of hormone levels. Approximately half of the transcription factors have multiple target genes (e.g., FOXK1, TP53 RORC, TCF7L1, ARID3A), whereas most of the remaining transcription factors have two to three target genes. (PBX1, SRF, NFYC, and TCF12). Most differentially expressed target genes had multiple regulatory transcription factors. Several differentially expressed genes have transcription factor functions, including *Gata3, Foxa2,* and *Tfap2b*.

## Discussion

Studies of targeted gene expression profiles indicate that the hypothalamus mediates the effects of prenatal inflammatory challenges on social, food intake, reproductive behaviors, and sensitivity to environmental stimuli [7]. Postnatal conditions such as stressors and sex contribute to variations in the effects of MIA on the hypothalamus. Mice exposed to PolyI:C-elicited MIA presented changes in pain sensitivity and reproduction behavior, and Gnrh1 was over-expressed in the hypothalamus of MIA males while under-expressed in MIA females [7]. Likewise, PRRSV-elicited MIA in pigs was associated with changes in physical activity, social behaviors, and amygdala transcriptome [1, 3, 28]. A study of the genes and pathways in the pig hypothalamus impacted by MIA and the stress of nursing withdrawal across sexes advance the understanding of how this brain structure can mediate the effects of prenatal and postnatal challenges on behavior and sensitivity to stimuli.

Among the 18,295 genes analyzed, changes in the profile of 222 genes (FDR-adjusted p value < 0.05) were associated with the effects of MIA, nursing withdrawal, sex, and interactions (S1 Table). Four-fold was the average fold change among the genes presenting sex or nursing effects alone or in interactions, and the profile was characterized by over-expression in pigs withdrawn from nursing and in males relative to nursed pigs or females. The observed over-expression of hypothalamic genes in response to the stress of weaning is consistent with reports of associated increases in the level of plasma cortisol and expression of genes Fos proto-oncogene, AP-1 transcription factor subunit, and *Crh* in the hypothalamus of 28-day-old pigs weaned relative to nursed [10]. The hypothalamus produces CRH that signals the pituitary gland to release adrenocorticotropin hormone (ACTH), which signals the adrenal glands to secrete cortisol hormones into the blood. In the present study, nursing withdrawal was associated with a 1.4 fold increase in the expression of *Crh*, and the level of cortisol was significantly higher in the blood of 21-day-old weaned relative to nursed pigs [29].

The fold change of genes differentially expressed in the MIA relative to the control group was, on average 6.5 units, indicating that prenatal inflammatory signals are associated with long-lasting gene over-expression in the hypothalamus. Consistent with these findings, most genes in the terpenoid biosynthesis pathway were over-expressed in the hippocampus of nursed male pigs exposed to PRRSV-elicited MIA, and terpenoids have anti-inflammatory effects [16]. The prenatal stressor of maternal anxiety during gestation was associated with enhanced HPA axis activity (characterized by high plasma cortisol levels) and depression symptoms in the offspring during adolescence [30]. Also, adult male guinea pigs born to females exposed to prenatal stress of strobe light during the rapid fetal brain growth phase presented higher basal plasma cortisol levels [31].

In the present study, MIA-exposed males had higher (albeit non-significant) expression of the cortisol-release precursor *Crh* relative to other groups. Prenatal immune activation interacting with the remaining factors studied, or alone, accounted for 103 genes. Similarly, nursing withdrawal interacting with MIA and sex or alone accounted for 85 differentially expressed genes. These results highlight the importance of studying the effects of prenatal and postnatal challenges in both sexes.

### Hypothalamic biological processes and genes networks associated with challenges and sex effects

Many differentially expressed genes in the study are annotated to processes related to the regulation of hormone levels (GO:0010817) (Table 1). This result is supported by the enrichment of processes related to G protein-coupled receptor signaling and activity and response to endogenous stimulus (GO:0007186, GO:0010469, and GO:0071495, respectively). Aligned

with our results, a study of transcriptome profiles in patients diagnosed with bipolar disorder, a MIA-associated phenotype, reported enrichment of the regulation of hormone levels process [32]. Also, the levels of transcript isoforms of the thyrotropin-releasing hormone were impacted by PRRSV-elicited MIA in the hippocampus of pigs [14]. The enrichment of positive regulation transcription by RNA polymerase II (GO:0045944) is consistent with reports that hormones modulate RNA polymerase II density patterns in the promoter and gene regions [33]. Growth, thyroid, and estrogen hormones have been associated with the recruitment of RNA polymerase II to regulatory regions [34].

Many genes annotated to the enriched hormone level and receptor signaling regulation processes shared profiles in response to MIA, nursing withdrawal, and sex effects (Table 2). These patterns included over-expression in the hypothalamus of pigs exposed to both challenges relative to a single challenge, and this profile was observed in *Agrp*, *Cga*, *En2*, *Fshb*, *Gata3*, *Pou4f1*, and *Ttr*. Another pattern is over-expression in MIA females relative to males, and this pattern was observed in genes including *Gh1*, *Pax3*, *Pou4f1*, *Prl*, and *Tfap2b*. Genes over-expressed in nursing withdrawn males relative to females included *Aldh1a1*, *Cxcl9*, *En2*, *Foxa2*, *Gh1*, *Pou4f1*, whereas the opposite pattern was detected for *Agrp*, *Cga*, *Kiss1*, *Nmu*, *Npy*, *Nr5a2*, and *Tshr*. The modification of the previous gene profiles can impact physiological processes because, for example, changes in the hypothalamic expression patterns of glucocorticoid receptors such as *Nr5a2* and *Ttr* can influence the modulation of the HPA axis. The under-expression of glucocorticoid receptor genes may disrupt the response to glucocorticoids, leading to increased ACTH release, production of adrenal steroids, and corticoid activity, influencing physiological processes such as puberty [7].

The detected profiles of genes associated with prenatal and postnatal challenge alone or in combination are aligned with published reports. Consistent with the over-expression of *Fshb* in the hypothalamus of pigs exposed to prenatal and postnatal challenges, over-expression of *Fshb*, gonadotropin-releasing hormone receptor (*Gnrhr*), and *Prl* was detected in the cerebellum of adult male mice that were neonatally inoculated with Thimerosal, an inflammatory agent associated with neurodevelopmental disorders [35]. The correlated profile of *Fshb* and *Lhb* observed in the present study is aligned with the inflammatory response to intraperitoneal LPS response in adult male mice that were associated with activation of the hypothalamus, disruption of the HPA axis, and elevated blood levels of FSH and LH [36]. The higher level of *Nmu* in MIA and nursing withdrawal groups agrees with the known physiological role of endogenous *NMU* in adaptation to environmental stimulus, inducing stress response after central administration [37]. Also, *Nmu* knockout mice have lower plasma corticosterone levels and lower stress indicators than wild-type mice [38].

Consistent with the observed under-expression of *Aldh1a1* in animals exposed to either challenge, this gene was under-expressed in the amygdala of neonatal rats exposed to early-life trauma modeled using odor-shock conditioning [39]. Likewise, *Aldh1a1* was under-expressed in the amygdala of 21-day-old pigs exposed to PRRSV-induced MIA [28]. The lower expression of *Aldh1a1* in the prenatal brain of mice exposed to MIA elicited by influenza, PolyI:C, and interleukin-6 has suggested that this gene is associated with aberrant neural development [40]. Also, decreased expression of *Aldh1a1* in the ventral tegmental area of the mice at PD22 was associated with repeated early exposure to social stress [41].

The higher level of *Gata3* in the hypothalamus of pigs exposed to MIA and postnatal stress agrees with the higher abundance of the protein *GATA3* in the brain of a mouse line that models autism spectrum disorder phenotypes [42]. A study of the effects of postnatal stress reported that the hypothalamus of female macaques exposed to housing in a novel room with unfamiliar conspecifics had increased expression levels of *Kiss1* relative to controls [43].

The sex-dependent effects of prenatal inflammation and nursing withdrawal on several genes detected in the present study are supported by previous findings. For example, the level of *FSHB* in the pituitary gland in response to chronic stress exposure modeled by regular handling was higher in males relative to female mice [44]. The hypothalamus of mice exposed to PolyI:C-caused MIA had increased expression of *Kiss1* relative to control mice, and the differential expression at P32 was higher in males than in females [7]. The previous finding led to the hypothesis of the peripubertal control that *Kiss1* exerts on abnormal brain function after prenatal immune challenge. The significant sex-dependent MIA effect characterized by under-expression of *Tfap2b* in the hypothalamus of MIA males relative to females is consistent with reports of under-expression in the cerebellum of male offspring of PolyI:C-treated female mice during gestation [45]. Also, male mice from a *Tfap2b* knockout line had increased susceptibility to stress [46].

The comparison of the hypothalamic gene networks from females and males exposed to prenatal inflammation relative to controls in the absence of a postnatal stressor indicates the over-expression of many interconnected genes (e.g., *Lhb*, *Fshb*, *Cga*) annotated to the regulation of hormone levels process (Fig 2). On the other hand, exposure to nursing withdrawal challenge reverted the profiles, and many genes were under-expressed in animals exposed to MIA relative to controls (e.g., *Gata3*, *Tfap2b*). The previous *Cga* pattern follows reports that LPS-elicited inflammation increases the expression of hypothalamic *Cga* in mice [47]. The over-expression of *Tfap2b* in male nursing pigs exposed to MIA was also reported in a brain transcriptomic study of mice offspring of dams exposed to the stressor of predator scent during gestation [48].

The network of genes that participates in the regulation of the hormone level process, many of which are also annotated to other enriched processes such as regulation of transcription by RNA polymerase II and G-protein-coupled receptor signaling pathway, highlighted the under-expression of many interacting genes in the withdrawn relative to nursing groups in offspring exposed to MIA (e.g., *Gata3*, *Lhb*, *Fshb*, *Cga*) (Fig 3). No predominant profile was detected in response to postnatal stress among control individuals not exposed to MIA. Consistent with the observed patterns of extreme under-expression of *Lhb* in nursing withdrawal relative to nursing groups exposed to MIA, neonatal LPS-induced inflammation furthers the suppression of luteinizing hormone pulse in adult rats [49]. The under-expression of *Lhb* is withdrawn relative to the nursing group was consistent across prenatal conditions and sexes, whereas *Fshb* was over-expressed in nursing withdrawal conditions in control females and males. The *Fshb* pattern is aligned with reports that the postnatal stress of social isolation is associated with lower LH and higher FSH secretion in rats [50].

The pattern of *Ttr* is characterized by under-expression in the hypothalamus of pigs exposed to either one challenge in the absence of the other challenge. The previous patterns are aligned with the under-expression of *Ttr* in the amygdala of 21-day-old pigs exposed to PRRSV -induced MIA [28] and in mice exposed to PolyI:C-induced MIA, both in the nucleus accumbens of [51] and the hippocampus [52]. The detection of sex effects and sex-dependent prenatal and postnatal challenge effects on the patterns of genes annotated to the regulation of hormone levels process has been previously reported. In mice, LPS-induced MIA was associated with deceleration of the migration of GnRH neurons to the forebrain in mice [4]. The previous findings supported the conclusion that LPS-initiated MIA can modulate the synthesis of GnRH and LH, leading to distinct abnormalities in testicular and ovarian development [53]. Also, LPS-initiated MIA in rats was associated with impaired sexual behavior in male offspring [2].

## Gene set and regulatory enrichment analysis of differentially expressed genes in the hypothalamus

The individualized GSEA study of the KEGG pathways and GO biological processes enriched for MIA, nursing withdrawal, or sex effects offered a complementary understanding of the specific mechanisms impacted by each factor. The over-representation of genes annotated to the neurotransmitter and inflammatory response categories was characterized by gene over-expression in individuals that experienced the prenatal or postnatal challenge. The prolonged disruption of the expression profile of genes associated with prenatal exposure to inflammatory signals and interaction with the postnatal stress of nursing withdrawal detected for the Influenza A and *Staphylococcus aureus* infection pathways (Table 3) can exemplify the double-hit hypothesis scenario [54].

The predominance of MIA-impacted genes that are annotated to immune response KEGG pathways such as influenza, allograft rejection, and antigen processing and presentation is consistent with reports of the enrichment of related functional categories in the amygdala of pigs and brains of mice exposed to viral and viral mimetic-elicited MIA [55, 56]. Likewise, the enrichment of immune-related functional categories among genes presenting sex-dependent MIA effects was reported in the hippocampus of pigs born to mothers exposed to virally-induced MIA and in the fetal cortex of mice exposed to PolyI:C-induced MIA during gestation [16, 57].

The over-representation of genes annotated to neurotransmitter regulation and transport processes that are affected by prenatal challenge interacting with postnatal challenge and sex is aligned with reports that MIA disrupts the development of neurotransmission in dopaminergic pathways underlying schizophrenia spectrum disorder phenotypes [58]. Neurotransmitters are commonly found in neurons alongside neuropeptides [59] and differential expression of neuropeptide genes was detected in the hippocampus of 60-day-old pigs exposed to PRRSV-elicited MIA [27]. Also aligned with the present findings, a study of MIA in mice detected a higher inhibitory synaptic tone of males and females, whereas changes in the excitatory synapse transmission were only detected in females relative to controls [60]. The between-sex differences in the prenatal inflammation effect further suggest the existence of sexual dimorphism in the synaptic pruning and neuronal disruption associated with MIA.

A noticeable finding is the enrichment of the olfactory transduction pathway, supported by olfactory receptor genes and the profile of most annotated genes opposite to that of the genes in the immune response and neurotransmitter pathways. Genes annotated to the olfactory transduction pathway include olfactory receptors (e.g., *51A4*, *52K1*, *51S1*, *2C1*, *51G1*, *51G2*, and *52R1*), which are over-expressed in individuals exposed to MIA, and in particular males and those exposed to nursing withdrawal. The detected participation of olfactory pathway genes agrees with reports that LPS-elicited MIA diminished the olfactory perception of neonatal and adult male and female rats and that these changes were not associated with neuroinflammation or changes in maternal care [61]. Likewise, our finding about the impact of nursing withdrawal stress on olfactory perception-related genes is consistent with reports relating chronic stress to disruption in the prefrontal cortex and hippocampus mechanisms and pathologies of the olfactory neural system [62].

The analysis of genes impacted by the effects of MIA, nursing withdrawal, and sex as targets of transcription regulation identified 80 enriched regulatory motifs and genomic tracks corresponding to 57 unique transcription factors (Table 4). Focusing on the genes related to hormone regulation of levels and signaling demonstrated that approximately half the transcription factors regulate the expression of a few genes, and the remaining factors target multiple genes.

The transcription factor SUZ12 is an example of a regulator of multiple differentially expressed genes in response to prenatal and postnatal stressors. A transcriptome study of the basolateral amygdala of male mice exposed to the chronic stress of repeated housing with an aggressive mouse identified the enrichment of SUZ12 upstream of differentially expressed genes [63]. SUZ12 was also enriched among genes in the hippocampus of 60-day-old pigs exposed to PRRSV-elicited MIA [27]. The present results are aligned with reports that mutations in *Suz12* have been associated with MIA-related autism spectrum disorder phenotypes [64]. The association of many differentially expressed genes to TCF7L1 is supported by a meta-analysis of transcriptome experiments that associated this transcription factor with autism spectrum disorder [65].

In addition to differentially expressed genes that are the target of shared transcription factors, several transcription factor genes were differentially expressed in response to MIA, nursing withdrawal, and sex. The transcription factors *Gata3*, *Foxa2*, and *Tfap2b* share the profile of over-expression in single challenge conditions (i.e., MIA or nursing withdrawal) and over-expressed in females exposed to MIA. Many detected transcription factors target genes annotated to inflammatory processes. The enriched transcription factor ALX4 has been associated with the activity of several inflammatory and stress factors, decreasing serotonin availability and increasing neuroinflammation and aberrant neuronal signaling [66]. The enrichment of the transcription factor Sirt6 is associated with differentially expressed genes in immune response pathways and relates to the role of SIRT6 in ameliorating neuroinflammation [67]. Also notable is that the over-represented transcription factor ARID3A inhibits the replication of the PRRSV in vivo and in vitro, and this is the virus used in the present study to elicit maternal inflammation [68].

Among the enriched transcription factors that target differentially expressed genes that participate in hormone regulation and signaling, CTBP2 is involved in suppressing interleukin-2 gene activation in T cells [69], and CTBPs can transactivate proinflammatory genes in the microglia and astrocytes that respond to LPS inflammatory challenge [70]. While the present study detected the enrichment of RORC in association with genes in the hormone signaling process, this transcription factor also participated in immune response processes. The ablation of RORC in mice lowers inflammation in autoimmune encephalomyelitis and lowers Th17 cells in the central nervous system that promote depression-like behaviors [71]. Similarly, depletion of the enriched transcription factor SRF during early life contributes to autism-like deficits in social interaction in adulthood and neurodevelopmental disorders [72]. The enrichment of PBX1, which targets the genes *Cga* and *Prl*, can be related to findings that this transcription factor was associated with autism spectrum disorder in a transcriptome study of adult cortical tissue in humans [73].

## Conclusions

An investigation of the genes and pathways influenced by MIA and postnatal stress enhanced our knowledge of the hypothalamus's intermediate role in modulating the impact of the factors studied on behavior and stimuli sensitivity. An unexpected finding is that prenatal and postnatal challenges interacting with sex significantly impacted genes annotated to the regulation of hormone levels and related pathways. The genes in the previous pathways tended to be over-expressed in individuals exposed to both challenges relative to individuals exposed to either challenge. Also, most genes were over-expressed in MIA females relative to males across nursing levels. These gene profiles suggest that environmental challenges before puberty can impact reproduction-related processes.

Notable, network analysis of genes in the regulation of hormone levels detected over-expression in nursed MIA relative to control pigs across sexes and a reversal of this trend within nursing withdrawn pigs. On the other hand, most genes were under-expressed in MIA withdrawn relative to nursing pigs. These findings exemplify the plasticity of the genes to accommodate either one or both challenges studied. In addition to endocrine signaling, neurotransmitter activity, and inflammatory response are molecular mechanisms impacted by prenatal and postnatal challenge and sex. Genes annotated to neurotransmitter and inflammatory response categories were overexpressed in the hypothalamus of pigs exposed to prenatal or postnatal challenges. Opposite to the immune response and neurotransmitter pathway genes, the olfactory transduction pathway genes were characterized by gene over-expression in weaned MIA males relative to other groups. Several transcription factors, including ALX4, *FOXA2*, SIRT6, and TP53, potentially target the differentially expressed genes and join the effect of the differentially expressed transcription factor genes *Gata3*, *Tfap2b*, and *Foxa2*. Altogether our findings showcase how two challenges or hits can interact in altering the hypothalamus molecular mechanisms associated with hormonal, immune response, and neurotransmitter processes.

## Supporting information

**S1 Fig. Experimental design scheme.**
(XLSX)

**S1 Table. Genes differentially expressed at False Discovery Rate-adjusted p value $< 0.05$ and log2(fold change) (LFC) across the effects of maternal immune inflammation (MIA), nursing withdrawal (Nur), sex and interactions (-).**
(XLSX)

**S2 Table. Enriched Gene Ontology biological processes including at least 5 differentially expressed (False Discovery Rate-adjusted p value $< 0.05$) genes across MIA, nursing withdrawal, sex, and interactions levels.**
(XLSX)

**S3 Table. Gene Ontology biological processes and KEGG pathways enriched at False Discovery Rate-adjusted p value $< 0.05$ for maternal immune activation (MIA), nursing withdrawal (Nur), sex, and interactions (-).**
(XLSX)

**S4 Table. The log2(fold changes) for each contrast between groups depicted in the gene networks.**
(XLSX)

**S5 Table. Enriched motifs, tracks, and transcription factors (normalized enrichment scores NES $> 3.5$) among genes presenting differential expression (False Discovery Rate FDR-adjusted p value $< 0.05$) for the effects of maternal immune activation, nursing withdrawal, sex, and interaction.**
(XLSX)

## Acknowledgments

The contribution of B. Southey in data management, organization, and formatting according to the National Center for Biotechnology Gene Expression Omnibus repository requirements is greatly appreciated.

## Author Contributions

**Conceptualization:** Sandra L. Rodriguez-Zas, Romana A. Nowak, Rodney W. Johnson.

**Data curation:** Nicole L. Southey.

**Formal analysis:** Sandra L. Rodriguez-Zas.

**Funding acquisition:** Sandra L. Rodriguez-Zas, Rodney W. Johnson.

**Investigation:** Nicole L. Southey, Laurie Rund, Adrienne M. Antonson.

**Methodology:** Sandra L. Rodriguez-Zas, Laurie Rund, Adrienne M. Antonson, Romana A. Nowak.

**Project administration:** Sandra L. Rodriguez-Zas, Rodney W. Johnson.

**Resources:** Sandra L. Rodriguez-Zas, Rodney W. Johnson.

**Supervision:** Sandra L. Rodriguez-Zas, Laurie Rund, Romana A. Nowak, Rodney W. Johnson.

**Visualization:** Nicole L. Southey.

**Writing – original draft:** Sandra L. Rodriguez-Zas, Nicole L. Southey.

**Writing – review & editing:** Sandra L. Rodriguez-Zas, Nicole L. Southey, Laurie Rund, Adrienne M. Antonson, Romana A. Nowak, Rodney W. Johnson.

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
