## [Decision Letter · Decision Letter 0]

5 Sep 2023

PONE-D-23-19245Prenatal and postnatal challenges affect the hypothalamic molecular pathways that regulate hormonal levelsPLOS ONE

Dear Dr. Rodriguez-Zas,

Thank you for submitting your manuscript to PLOS ONE. After careful consideration, we feel that it has merit but does not fully meet PLOS ONE’s publication criteria as it currently stands. Therefore, we invite you to submit a revised version of the manuscript that addresses the points raised during the review process.

We look forward to receiving your revised manuscript.

Kind regards,

Prof. Dr. Dragan Hrncic, MD, PhD

Academic Editor

PLOS ONE

Journal Requirements:

2. Please expand the acronym “USDA, NIFA, AFRI, ILLU” (as indicated in your financial disclosure) so that it states the name of your funders in full. This information should be included in your cover letter; we will change the online submission form on your behalf.

"This research was funded by USDA NIFA AFRI award number 2018-67015-27413, USDA NIFA ILLU award number 2022-38420-38610, and NIH NIDA award number P30 DA018310."

**Additional Editor Comments:**

- please do respond in detail to reviewer comments and directions to improve your manuscript 

Reviewers' comments:

Reviewer's Responses to Questions

**Comments to the Author**

1. Is the manuscript technically sound, and do the data support the conclusions?

Reviewer #1: Partly

2. Has the statistical analysis been performed appropriately and rigorously? 

Reviewer #1: Yes

3. Have the authors made all data underlying the findings in their manuscript fully available?

Reviewer #1: Yes

4. Is the manuscript presented in an intelligible fashion and written in standard English?

Reviewer #1: Yes

5. Review Comments to the Author

Reviewer #1: This work used RNA sequencing to study how prenatal challenge with maternal immune activation (MIA) and postnatal challenge with nursing impact the molecular pathways of hypothalamic pathways, and examine the effects are different among female and male pigs. The main concerns are as follows,

1) Instead of whole-transcriptome sequencing, the work used a target panel of 222 genes. Why? How these genes were chosen? How this limitation will affect the functional analysis of differentially expressed genes (DEGs)?

2) The main result is to present the analysis results of DEGs across three factors, functional significance and the possible regulation mechanism. The current form is hard to follow. The authors may consider to reorganize the results in a more intuitive way.

3) Main result is focused on the hormonal regulation pathway. However, a lot of discussion is on the olfactory transduction pathway while only one enrichment result was presented about the pathway.

4) There is no confirmation from the protein levels on the findings.

Minor points:

1) Figure 1 is hard to digest. Only the DEGs are included and the box-whisker plot seems inappropriate since DEGs are either upregulated (logFC >0) or downregulated (logFC<0). Non-DEGs should have a logFC close to 0.

2) How are the expression levels of 222 genes quantified? Esp in the comparisons between Ctrl and MIA, and between Wea and Nur, the logFCs of DEGs were biased towards the negative side and the positive side, respectively?

3) “Sex” is a different factor from other two. “prenatal challenge with sex” and other similar descriptions are confusing and misleading. The authors may consider rewrite the description on the sex factor.

6. PLOS authors have the option to publish the peer review history of their article (what does this mean?). If published, this will include your full peer review and any attached files.

Reviewer #1: No

---

## [Author Response · Author response to Decision Letter 0]

8 Sep 2023

September 7, 2023

Reply to the editorial and reviewers’ recommendations on the manuscript PONE-D-23-19245

Dear Professor Dr. Hrncic and reviewer,

We appreciate for the opportunity to revise our manuscript PONE-D-23-19245. We have modified the submission to address questions and integrate the recommendations offered to us. The suggestions have strengthened our manuscript, and we thank the reviewers for their feedback. 

Below is the list of recommendations made by the editorial team and reviewer, and in our point-by-point reply, we describe the manuscript modifications that address the suggestions. 

Thank you

Sandra Rodriguez Zas

Reviewer #1: This work used RNA sequencing to study how prenatal challenge with maternal immune activation (MIA) and postnatal challenge with nursing impact the molecular pathways of hypothalamic pathways, and examine the effects are different among female and male pigs. The main concerns are as follows,

1) Instead of whole-transcriptome sequencing, the work used a target panel of 222 genes. Why? How these genes were chosen? How this limitation will affect the functional analysis of differentially expressed genes (DEGs)?

REPLY. Agree. Clarifications were added in page 6 and in page 15. A whole-transcriptome analysis was undertaken. Please refer to page 5, second paragraph about the “test the effects of prenatal inflammation, nursing withdrawal, sex, and interactions on 18,295 genes”. The reference to 222 genes in the Discussion correspond to the number of genes that presented differential expression at FDR-adjusted p value < 0.05 (page 15 first paragraph). 

2) The main result is to present the analysis results of DEGs across three factors, functional significance and the possible regulation mechanism. The current form is hard to follow. The authors may consider to reorganize the results in a more intuitive way.

REPLY. We agree in that the flow of result presentation must be amenable to readers while meeting scientific standards. The order of result presentation follows statistical and biological better practices. Following the logical approach of deductive reasoning, the progress of the result presentation goes from functional categories, corresponding genes and interactions impacted by the factors studied (Tables 1 and 2 and Figs 2-3). Subsequently, the presentation of results focuses up-regulated or down-regulated functional categories (Table 4) and ends with transcription factors. Also consistent with better practices, results within a table were obtained from a common model, method, and specifications.

3) Main result is focused on the hormonal regulation pathway. However, a lot of discussion is on the olfactory transduction pathway while only one enrichment result was presented about the pathway.

REPLY. Agree. Clarifications were added in page 10, and 18. The network of genes in the hormonal regulation pathway was selected for visualization across 12 contrasts because of the significant enrichment, and because it includes genes from other enriched pathways including regulation of transcription by RNA polymerase II, G-protein coupled receptor signaling pathway, cellular response to endogenous stimulus, regulation of signaling receptor activity. Also, the number of genes annotated to the hormonal regulatory pathway is amenable to interpretable visualization. The larger number of genes annotated to the olfactory transduction pathway resulted in a denser network that challenged accessible visualization and interpretation for the 12 group comparisons reported. 

4) There is no confirmation from the protein levels on the findings.

REPLY. Agree. A clarification was added in page 6. We agree with the implementation of strategies to minimize false positive results, and in the unbiased validation of results whenever possible. The large number of samples analyzed (72 individuals) and significant genes across effects and contrasts prevents an unbiased confirmation of all group contrasts and 220 genes discussed using protein-based analysis. The replicability of the findings is supported by the conservative criteria in the analysis, and genes and categories selected for interpretation (i.e., simultaneous FDR adjusted p-value < 0.05). Also, genes with low sequence reads in any one group were not reported (Materials and methods section). To convey high degree of robustness in the findings, our discussion highlighted differentially expressed genes that were both supported by enriched functional categories and have been previously associated with maternal immune activation, stress, and sex differences as evidenced by our thorough reference list. 

Minor points:

1) Figure 1 is hard to digest. Only the DEGs are included and the box-whisker plot seems inappropriate since DEGs are either upregulated (logFC >0) or downregulated (logFC<0). Non-DEGs should have a logFC close to 0.

REPLY. Agree. A clarification was added in page 6. We agree that have Fig 1 depicted all 18,295 genes analyzed, the box-whisker plots would have been centered around 0. Fig 1 focuses on the differentially expressed genes and therefore the plots will not necessarily be centered around 0. The plots aim at highlighting that the effects studied do not have a balanced effect in up-regulating some genes and down-regulating in similar magnitude or quantity. 

2) How are the expression levels of 222 genes quantified? Esp in the comparisons between Ctrl and MIA, and between Wea and Nur, the logFCs of DEGs were biased towards the negative side and the positive side, respectively?

REPLY. Agree. Clarifications were added in pages 5 and 6. We agree that the description of the quantification is important in assessing the findings. A generalized linear model was used to test the effects of prenatal inflammation, nursing withdrawal, sex, and interactions on 18,295 genes analysis. Among these, some genes were differentially expressed at FDR-adjusted p value < 0.05 between MIA and Ctrl and others were differentially expressed between Wea and Nur. Fig 1 demonstrates that most of the impacted genes were over-expressed in individuals exposed to MIA and in individuals exposed to stress.

3) “Sex” is a different factor from other two. “prenatal challenge with sex” and other similar descriptions are confusing and misleading. The authors may consider rewrite the description on the sex factor.

REPLY. Agree. Clarifications were added in the abstract and in page 8. We agree that while the prenatal and postnatal challenges are distinct in nature to sex differences. Following standard statistical approaches, we tested the effect of each one of the factors (maternal immune activation, stress, and sex) and also their interaction. These interactions aim at identifying whether the effect of maternal immune activation (or stress) depends or varies with sex. In consideration that 79 genes presented significant interaction of either prenatal or postnatal challenge with sex, we favor reporting these interactions in paragraphs proximal to interpretations that focus on the challenges to remind readers of the sex-dependent nature of some of the profiles detected.

---

## [Decision Letter · Decision Letter 1]

25 Sep 2023

PONE-D-23-19245R1Prenatal and postnatal challenges affect the hypothalamic molecular pathways that regulate hormonal levelsPLOS ONE

Dear Dr. Rodriguez-Zas,

Thank you for submitting your manuscript to PLOS ONE. After careful consideration, we feel that it has merit but does not fully meet PLOS ONE’s publication criteria as it currently stands. Therefore, we invite you to submit a revised version of the manuscript that addresses the points raised during the review process.

We look forward to receiving your revised manuscript.

Kind regards,

Prof. Dr. Dragan Hrncic, MD, PhD

Academic Editor

PLOS ONE

Journal Requirements:

Reviewers' comments:

Reviewer's Responses to Questions

**Comments to the Author**

1. If the authors have adequately addressed your comments raised in a previous round of review and you feel that this manuscript is now acceptable for publication, you may indicate that here to bypass the “Comments to the Author” section, enter your conflict of interest statement in the “Confidential to Editor” section, and submit your "Accept" recommendation.

Reviewer #1: (No Response)

2. Is the manuscript technically sound, and do the data support the conclusions?

Reviewer #1: Yes

3. Has the statistical analysis been performed appropriately and rigorously? 

Reviewer #1: Yes

4. Have the authors made all data underlying the findings in their manuscript fully available?

Reviewer #1: Yes

5. Is the manuscript presented in an intelligible fashion and written in standard English?

Reviewer #1: Yes

6. Review Comments to the Author

Reviewer #1: Figure 1 remains the same which requires a change of presentation. The box-whisker plot is inappropriate to present the distribution of two different subsets (up and down DEGs here). It is advised to use two separate box-whisker plots (one for up and one for down) or two separate density plots. The authors may also consider to add a density distribution plot of the normalized counts to explain why the distribution of DEGs are skewed in the first 2 contrasts.

7. PLOS authors have the option to publish the peer review history of their article (what does this mean?). If published, this will include your full peer review and any attached files.

Reviewer #1: No

---

## [Author Response · Author response to Decision Letter 1]

28 Sep 2023

Sep 27 2023

Reply to the reviewers’ and editorial recommendations on the manuscript PONE-D-23-19245R1

Dear Professor Dr. Hrncic and reviewers,

Once again, we express our appreciation for the opportunity to revise our submission, integrating the recommendations offered to us. 

Below is the reply to the feedback. 

Thank you

Sandra Rodriguez Zas and co-authors

Review Comments to the Author

Reviewer #1: Figure 1 remains the same which requires a change of presentation. The box-whisker plot is inappropriate to present the distribution of two different subsets (up and down DEGs here). It is advised to use two separate box-whisker plots (one for up and one for down) or two separate density plots. The authors may also consider to add a density distribution plot of the normalized counts to explain why the distribution of DEGs are skewed in the first 2 contrasts. The title of the figure and in-text reference were revised in the manuscript accordingly.

REPLY. Agree. Figure 1 has been revised. I apologize for failing to understand the original comment. Following the reviewer’s suggestion, two separate box-whisker plots (one for up and one for down) were created for each effect. Other studies of maternal immune activation are reporting uneven distribution of positive and negative fold changes among the differentially expressed genes (e.g., Kalish, B.T., Kim, E., Finander, B. et al. Maternal immune activation in mice disrupts proteostasis in the fetal brain. Nat Neurosci 24, 204–213 (2021). https://doi.org/10.1038/s41593-020-00762-9; Cesar P CanalesMyka L EstesKarol CichewiczKartik AngaraJohn Paul AboubecharaScott CameronKathryn PrendergastLinda Su-FeherIva ZdilarEllie J KreunEmma C ConnollyJin Myeong SeoJack B GoonKathleen FarrellyTyler W StradleighDeborah van der ListLori HaapanenJudy Van de WaterDaniel VogtA Kimberley McAllisterAlex S Nord (2021) Sequential perturbations to mouse corticogenesis following in utero maternal immune activation eLife 10:e60100); Sarieva, K., Kagermeier, T., Khakipoor, S. et al. Human brain organoid model of maternal immune activation identifies radial glia cells as selectively vulnerable. Mol Psychiatry (2023). https://doi.org/10.1038/s41380-023-01997-1)

---

## [Editor Report · Decision Letter 2]

3 Oct 2023

Prenatal and postnatal challenges affect the hypothalamic molecular pathways that regulate hormonal levels

PONE-D-23-19245R2

Dear Dr. Rodriguez-Zas,

We’re pleased to inform you that your manuscript has been judged scientifically suitable for publication and will be formally accepted for publication once it meets all outstanding technical requirements.

Kind regards,

Prof. Dr. Dragan Hrncic, MD, PhD

Academic Editor

PLOS ONE
---

## [Editor Report · Acceptance letter]

8 Oct 2023

PONE-D-23-19245R2 

Prenatal and postnatal challenges affect the hypothalamic molecular pathways that regulate hormonal levels 

Dear Dr. Rodriguez-Zas:

I'm pleased to inform you that your manuscript has been deemed suitable for publication in PLOS ONE. Congratulations! Your manuscript is now with our production department. 

Kind regards, 

on behalf of

Professor Dragan Hrncic 

Academic Editor

PLOS ONE